# Investigation on Pollution Control Device (PCD) in iron foundry industry to reduce environmental chemicals

**Krishnaraj Ramaswamy**[1,2]*, **Leta Tesfaye Jule**[2,3], **Nagaprasad N.**[4], **Kumaran Subramanian**[5], **Shanmugam. R.**[6], **Priyanka Dwarampudi. L.**[7], **Venkatesh Seenivasan**[8]

**1** Mechanical Engineering Department, College of Engineering Science, Dambi Dollo University, Dambi Dollo, Ethiopia, **2** Centre for Excellence in Technology Transfer and Incubation, Dambi Dollo University, Dambi Dollo, Ethiopia, **3** Department of Physics, College of Natural and Computational Science, Dambi Dollo University, Dambi Dollo, Ethiopia, **4** Department of Mechanical Engineering, ULTRA College of Engineering and Technology, Madurai, Tamil Nadu, India, **5** Centre for Drug Discovery and Development, Sathyabama Institue of Science and Technology, Chennai, Tamil Nadu, India, **6** Department of Pharmacognosy, TIFAC, CORE-HD, JSS College of Pharmacy, JSS Academy of Higher Education & Research, Ooty, Tamil Nadu, India, **7** Department of Pharmacognosy, JSS College of Pharmacy, JSS Academy of Higher Education & Research, Ooty, Tamil Nadu, India, **8** Department of Mechanical Engineering, Sri Eshwar College of Engineering, Coimbatore, Tamil Nadu, India

* prof.dr.krishnaraj@dadu.edu.et

**Data Availability Statement:** All relevant data are within the paper and its Supporting information files.

## Abstract

Right from the olden days, many products have been made according to foundry practices in order to generate prosperity in the societies in which they operate while reaping these types of benefits through the operation of foundries. It is alarming that the emissions released by foundries affect human health. Therefore, foundries installed Pollution Control devices (PCDs), in accordance with this development; researchers examined the effectiveness of these PCDs in controlling emissions from foundries in different parts of the world. The emission control obtained by installing these PCDs is explained in this article based on the data gathered from the survey. The cartridge filter equipped with an induction furnace reduced the concentration of SPM to less than 20 mg/Nm$^3$. This result of the investigation indicates that the cartridge filter built into the induction furnace achieves the best efficiency in controlling contaminants from iron foundries. Interestingly, the operation of the cartridge filter has yet to be documented. Therefore, the construction operation, the performance of the cartridge filter, and its efficiency in achieving contaminations control in foundries are described. This will provide useful information on the use of cartridge filters in an induction furnace to reduce Iron foundry emissions.

## Introduction

Most of the products in the world are made using foundry practices [1–3]. Foundry practices include melting metals, pouring these molten metals into molds, solidifying these molten

**Funding:** The authors received no specific funding for this work.

**Competing interests:** The authors have declared that no competing interests exist.

metals in molds, removing solidified metals from molds, and cleaning these solidified metals [4]. These cleaned and solidified metals are called castings [5]. These practices are carried out in companies belonging to the foundry industry. The foundry industry provides products that are widely used in products such as pumps, automobiles, and compressors. Thus, the contribution of the foundry industry is vital and crucial to many economies [6]. However, emissions from the foundry industry cause pollution [7–9]. These emissions are mineral dust and organic carbon emitted during the melting, sand casting, and cleaning of castings [4, 10–13]. With the growing concern about environmental issues, the pollution problems caused by the foundry industry must be seriously and urgently investigated [9, 14–16].

Companies belonging to the foundry industry are more likely to come under public scrutiny, and therefore they are required to reduce emission levels by using pollution control Devices (PCDs). In this context, theorists and practitioners increasingly consider the need to install PCDs in the foundry industry [17–20]. In addition, European regulations have prompted the foundry industry to use PCDs [21–23]. Such regulations adopted in several countries have promoted the conduct of research in the foundry industry. The research described in this article was conducted against this background. The investigation reported in this article investigated pollution control using PCDs in iron foundries in the Tamilnadu State of India.

In Tamilnadu, a large number of companies produce products like wet grinders, pumps, and motors are largely manufactured. In order to cater to the needs of the iron castings used by these companies, the iron foundry industry has been existing as a major industry sector in Tamilnadu. An increasing number of iron foundry industries in Tamilnadu have improved the economy and quality of life of people. However, the pollutants emitted by these iron foundries affect the healthy living of people in Tamilnadu [16, 24]. Those pollutants are emitted due to poor production process, poor safety management, and low air pollution controlling practices employed while carrying out the foundry practices [25]. This has created a situation to install PCDs. As a result, some of the iron foundries situated in Tamilnadu have installed PCDs. This situation has revealed the need to study the impact of PCDs on controlling pollution in iron foundries located in the Tamilnadu State of India. In order to meet this research need, the study reported in this paper was carried out.

Section 2 presents the details of the literature review conducted during the study reported here. Section 3, the history of carrying out pollution control in India, is narrated. In the same section, it is reported that no studies have been done on pollution control in iron foundries in the Tamilnadu state of India and describes attempts to gather relevant information from foundries in the Tamilnadu state. In section 4, the collected information is analyzed, and the same section describes the construction and operation of the cartridge filter and its ability to control contamination in foundries. In the last section, the article concludes with reference to the possibilities of continuing future research towards pollution control through the use of PCDs.

## Pollution control in iron foundry: A review from literature arena

A literature search was conducted to find research papers describing pollution control in iron foundries using PCDs. It was surprising that only eight papers reporting this type of research could be compiled from leading databases, namely Science Direct, Emerald Insight, and Springer. In total, these documents related to pollution control in iron foundries in three categories. Statistics from these articles reporting research in these three categories are presented in Fig 1. As noted, two articles could each be grouped into categories 1 and 2. Four articles are grouped into category 3. These four articles have been reported by researchers who combine the topics covered in the articles into categories 1 and 2.

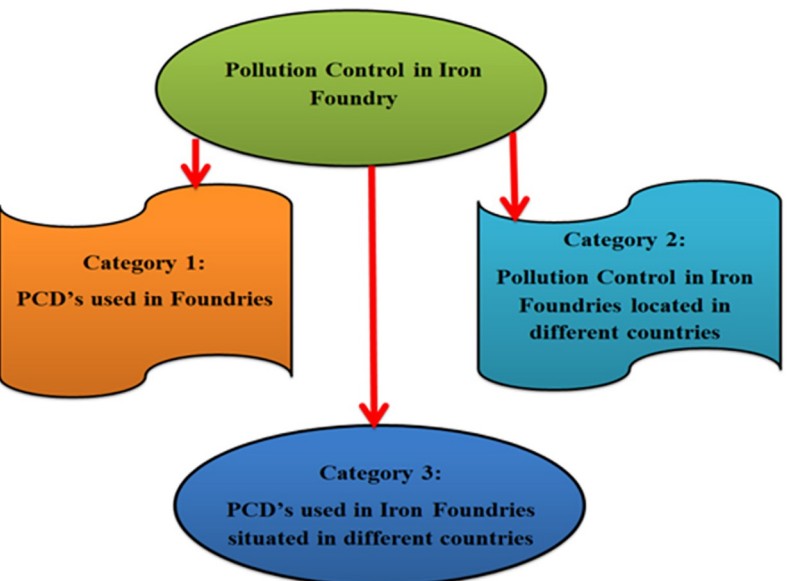

**Fig 1. Pollution control in iron foundry integration.**

The first category examines the model and devices used to control pollution in iron foundries. In the second category, the application of PCDs in iron foundries in different parts of the world is evaluated. As noted above, the third category of articles addresses issues addressed in both the first and second categories of articles. The information gathered from these three categories of articles is highlighted in the following subsections.

## PCDs used in foundries

During the review of the literature reported here, two articles have been identified that deal with devices used to control pollution in iron foundries. Information from these articles is highlighted here. Biswas et al. [26] discussed the reduction of exhaust emissions from the dome factory. These authors have listed some devices used in gas treatment plants connected to domes. Some of these devices are cyclones, draft fans, and chimney scrubbers. Peukert and Wadenpohl [27] described the construction and operation of the cyclone, electrostatic precipitator (ESP), wet scrubber, cloth filter, and high-temperature filtrates. These authors did not explicitly address pollution control in iron foundries. However, the devices they describe can be used to control the emissions of exhaust gases coming from iron foundries.

## Pollution control in iron foundries located in different countries

During the literature search reported here, two articles on the pollution caused by iron foundries in Italy and South Korea were overcome. Highlights of the information in these articles are presented here. Polizzi et al. [28] studied air pollution caused by aluminum and cast-iron foundries in the Turin district of Italy. These authors found that the concentration of iron and aluminum particles varied during different periods of the study. These authors also found that pollution from these foundries affected human lung function. Yu et al. [29] examined the smoke levels of polychlorinated dibenzo-p-dioxin and dibenzofurans (PCDD/F) emitted by metallurgical industries in South Korea. These authors estimated the levels of these pollutants but did not identify their practical implications.

## PCDs used in iron foundries situated in different countries

Some researchers have focused their research on the use of pollution control devices in foundries in various parts of the world. The information gathered from these documents is presented in this subsection. Pal et al. [4] investigated the use of PCDs in iron foundries that are located and housed in a dome. These authors used a model called a high-throughput dome (CBD) to reduce carbon monoxide formation in the dome. The use of DBC is encouraged in India by the Swiss Development Control Agency (SDC) and the Energy Resources Institute (TERI). The construction and operation of the DBC model are illustrated in this article. It also lists the devices available to clean the exhaust from the dome. These devices are centrifugal separators, low energy scrubbers, and high-intensity scrubbers. By applying the DBC, in the study described in this article, the Suspended Particulate Matter (SPM) has been reduced from 2000 mg/Nm$^3$ (milligrams per normal cubic meter) to a value below 70 mg/Nm$^3$. The installation of a Venturi wet scrubber resulted in the emission of SO$_2$ at only 40 mg/Nm$^3$.

Rabah [11] analyzed the profitability of installing PCDs in iron foundries in Egypt. PCDs listed in this article include cyclone, cloth filters, electro filters, and wet scrubbers. This author presented data on total particulate matter emissions (in mg/Nm$^3$) emitted from crucibles, short rotary, cupola, and electric induction furnaces from iron foundries in Egypt. After reviewing the economic considerations, this author concluded that wet scrubbing systems are the most suitable emission control equipment for iron foundry units in Egypt. Mukherjee [25] reported the scenario of foundries in the Howrah district of India. As Pal et al. [4] This author also reported that SDC and TERI were encouraging foundries in India to use a DBC equipped with a venturi scrubber to control the pollution caused by dome operations. This author described the methodology of pollution control in foundries using three devices, namely the cyclone, the scrubber, and the fabric filter.

Fatta et al. [30] have investigated the emissions from foundries in Cyprus. These authors presented several guidelines for preventing pollution and emissions from foundries. An interesting suggestion is that the induction furnace is preferable to the cupola furnace to control contamination from iron foundries. These authors studied the use of wet scrubbers and venturi scrubbers to reduce foundry emissions. Lv et al. [1] studied the release of PCDD / F from iron foundries in China. They said the anti-pollution devices used in Chinese foundries are the fabric filter, the cyclone, and the wet scrubber.

The literature review reported above indicates that significant efforts are being made to control exhaust fumes from iron foundries in various parts of the world. In carrying out these efforts, these foundries used devices.

From Table 1, it is evident that different researchers are concentrating on the reduction of particulate matter. It is evident from various research finding from the year 2004 to 2020 that

**Table 1. Particulate matter concentration emission review.**

| Sl.No | Pollution Control Device | Particulate Matter concentration (mg/Nm$^3$) | Reference |
|:---:|---|:---:|:---:|
| 1 | Cartridge filter | 20 | Current Paper |
| 2 | Wet scrubber | 38 | Kumar et al. (2020) |
| 3 | Cyclones | 50 | Guinot et al. (2016) |
| 4 | Venturi wet scrubber | 67.90 | Aksu et al. (2015) |
| 5 | Wetcap collectoer | 70 | Safar et al. (2010) |
| 6 | Wetcap collector | 70 | Pal et al. (2008) |
| 7 | Drycap collector | 107.90 | Pineda-Martinez et al. (2014) |
| 8 | Dry cap collector | 350 | Fatta et al. (2004) |

particulate matter concentration has reduced from 350 mg/Nm$^3$ to 38 mg/Nm$^3$ with different pollution control mechanisms such as Venturi scrubber, Cyclone, Wetscrubber, Wetcap collector, Dry cap collector as shown in Table 1. The current paper addresses the latest innovative technology of cartridge filters where the particulate emission is brought below 20 mg/Nm$^3$. This finding will be a new milestone for the researchers and practitioners to select appropriate control devices, which will make the foundry industries to function hassle-free.

## Methods and material

### Pollution control in India and the absence of studies on pollution control in iron foundries of Tamilnadu

India is one of the countries where efforts are being made to control pollution from iron foundries [4–9]. The pressure from the Indian government to combat pollution from industrial growth is also fueling the increased use of PCDs in iron foundries across different groups of foundries in other parts of India. Stockholm Conference on the Human Environment in 1972, the Indian government began enacting laws to combat pollution from rapid industrialization. One of these is the Air (Pollution Prevention and Control) Act 1981. Subsequently, the Indian government established the Central Pollution Control Board (CPCB) to promote pollution control standards. The oversight of pollution control activities has been entrusted to institutions controlled by the state governments of India. These institutions are known in India as the National Pollution Control Offices. These institutions are known as the Indian states in which they operate. For example, in the state of Tamilnadu in India, this institution is known as the Tamilnadu State Pollution Control Board (TSPCB). In addition to supervising the implementation of pollution control activities, the TSPCB also participates in the preparation of pollution reports generated by various industrial sectors. A striking observation made during the review of the literature reported in the previous section is that the investigation into the control of pollution caused by foundries in the Tamilnadu state of India has not yet been completed. Although [31] studied the heavy metal content in the ambient air in the Coimbatore district of India, no exclusive research was conducted on the use of PCDs in iron foundries in the same state. As part of developing these conclusions by conducting the literature review described in the previous section, the use of PCD in foundries in Tamilnadu state was explored.

### PCDS employed and their impact in the iron foundries of Coimbatore

The study reported here was conducted in two phases. Initially, PCDs used in foundries in Tamilnadu were studied. To conduct this study, it was decided to select a sample foundry that would reflect the situation in Tamilnadu foundry units with regard to decontamination aspects. In order to choose the foundries in Tamilnadu to collect relevant data, it was necessary to identify the area that is severely polluted with fine particles less than 1μm [32–34]. Only 25 iron foundries were found, which convenes the following criteria. i) The iron foundry unit can use a dome or induction furnace for melting metals [35]. (This criterion has been chosen because most foundries in India use dome or induction furnaces to melt metals) ii) the iron foundries can meet the TSPCB's audit requirements and can track the pollution emitted by the foundries. (This criterion has been chosen because collecting data from the files sent to the TSPCB would be very authentic). iii) The iron foundry must have a PCD installed to control the release of pollutants. (This criterion was chosen to study the impact of the use of PCDs in iron foundries on pollution control).

    After examining the above criteria, the two experts identified 24 iron foundries in Tamilnadu. The first author preferred to add a foundry to this list even though it is not located in

Tamilnadu. The reason this foundry has been included in the list is that it is a scientifically managed foundry and features a relatively new PCD called a cartridge filter. This careful selection allowed us to collect relevant and authenticated data on the use of PCDs in these 25 iron foundries. These data represented approximately 1500 iron foundries in Tamilnadu. In this regard, a datasheet has been prepared to collect relevant data on the use of PCDs. By interviewing the managers of these iron foundries [4–9], the data in this sheet is complete. In particular, the rejection rate, the GPS, and the pollutants before and after the implementation of the PCDs were entered into this sheet by questioning these officials. These operators provided the data by referring to pollution data periodically submitted by external pollution control bodies such as the TSPCB.

## Results and discussion

Out of the 25 iron foundries, the PCDs installed in four foundries were not in good condition. For example, data was collected on PCDs that were effectively used in 21 iron foundries. These data are presented in Table 1. Even in these 21 iron foundries, some data could not be collected. For example, in Iron Foundry 4, the discharge rate prior to the installation of the wet plug manifold was not available. As this table shows, a dome is used in ten iron foundries. An induction furnace is used in the remaining 11 iron foundries. PCDs used in iron foundries equipped with cupola furnaces include cyclones, venturi wet scrubbers, energy-efficient scrubbers such as spray towers or wet cap and dry cap collectors [4–9]. The PCDs used in iron foundries built into an induction furnace are the wet scrubber, the venturi wet scrubber, the bag filter, and the cartridge filter. An interesting observation is that so far, no researcher has treated the cartridge filter as a PCD used in iron foundries. Cartridge filter not found. Referring to the data presented in Table 2, the impact of PCDs used in Tamilnadu iron foundries is described in the following subsections.

### Impact of dry cap collector

As shown in Table 2, three of the 21 iron foundries surveyed installed a dry plug manifold. The dry cap collector is a kind of pollution control equipment mounted on the dome pole. A spark arrester is fitted to the top of the dome. There is a cap with a deflector around the spark arrester. This assembly is called a dry plug collector. When dusty air enters the dry cap, particles settle at the bottom of the dry cap. Baffles are installed at the bottom of the dry cap. The baffles act as a barrier to prevent dusty air from escaping into the atmosphere. Clean air enters the atmosphere through the chimney. As shown in Table 1, a dry cap manifold equipped with a dome was found to reduce SPM from 1750 mg/Nm$^3$ to 496 mg/Nm$^3$ in foundry-3, 394 mg/Nm$^3$ in foundry-7, and 366 mg/Nm$^3$ in the foundries-9. In these three iron foundries, the particle concentration turned out to be higher than the specification of the CPCB, which is 150 mg/Nm$^3$ [4–9].

### Impact of wet cap collector

As shown in Table 2, the wet cap manifold is installed in iron foundries 2, 4, 6, 8, and 10. The wet cap manifold is another type of contamination control equipment. The function of the wet hood collector is the same as that of the dry hood collector. The additional feature of the wet cap is that water is sprayed into the housing, which is mounted on top of the dome with a water bike line. The dusty air is cleaned at high speed by water, and the solid particles settle in the settling tank. The water is recycled using a pump. As shown in Table 1, in iron foundries 2, when the wet cap manifold is equipped with a dome, the SPM concentration is reduced from 2500 mg/Nm$^3$ to 110 mg/Nm$^3$. In iron foundries 4, 6, and 10, no particulate matter

**Table 2. PCDs installed in foundries of Tamilnadu state and their impact on pollution prevention.**

| Iron Foundry number | Furnace used | PCD | Stack temperature (average) in degree Kelvin | | Velocity (m/sec) | Discharge rate (liters/hour) | | SPM (mg/Nm³) (average) | | The concentration of pollutants in (mg/Nm³) | | | |
|---|---|---|---|---|---|---|---|---|---|---|---|---|---|
| | | | Before | After | | Before | After | Before | After | $SO_2$ | $NO_x$ | $CO_2$ | CO |
| | | | Installing the PCD | | | Installing the PCD | | Installing the PCD | | | | | |
| 1 | Cupola | Cyclone | 200 | 39 | 7.4 | 8350 | 4099 | 300 | 67 | 2.7 | 1.2 | - | - |
| 2 | Cupola | Wet cap Collector | 200 | 150 | - | 5670 | 4500 | 2500 | 110 | 10 | 17 | - | 0.2 |
| 3 | Cupola | Dry cap collector | 250 | 200 | - | - | - | 1750 | 496 | 340 | 18 | 4.5 | - |
| 4 | Cupola | Wet cap collector | - | 150 | 5.52 | - | 1060 | - | 110 | 31 | 21 | - | 0.2 |
| 5 | Cupola | Venturi wet Scrubber | - | 37 | 11.4 | - | 2043 | - | 51 | 26 | 11 | - | 0.2 |
| 6 | Cupola | Wet cap Collector | - | 150 | 11.28 | - | 1066 | - | 102 | 21 | 15 | - | - |
| 7 | Cupola | Drycap collector | - | 200 | - | - | - | - | 394 | 38 | 18 | - | - |
| 8 | Cupola | Wet cap Collector | - | 150 | 11.4 | - | 1757 | - | 95 | 66 | 20 | 5 | - |
| 9 | Cupola | Drycap collector | - | 200 | - | - | - | - | 366 | 63 | 47 | ND | - |
| 10 | Cupola | Wet cap collector | - | 160 | 6.2 | - | 1360 | - | 105 | 30 | 19 | ND | - |
| 11 | Induction | Venturi wet scrubber | - | 28 | 9.86 | - | 5588 | - | 31 | 19.2 | 3.1 | 5 | ND |
| 12 | Induction | Venturi wet scrubber | - | 29 | <10 | - | 4926 | - | 37 | 21 | 12 | ND | ND |
| 13 | Induction | Wet scrubber | - | 53 | 9.86 | - | 5588 | - | 52 | 19.2 | 31 | ND | 0.2 |
| 14 | Induction | Wet scrubber | - | 38 | 9.2 | - | 5260 | - | 51 | 12 | 7 | ND | ND |
| 15 | Induction | Wet scrubber | - | 32 | 10.2 | - | 4592 | - | 50 | 15 | 8 | ND | ND |
| 16 | Induction | Wet scrubber | - | 35 | 11 | - | 6120 | - | 51 | 12 | 6 | ND | ND |
| 17 | Induction | Wet scrubber | - | 26 | 9.8 | - | 4980 | - | 49 | 14 | 7 | ND | ND |
| 18 | Induction | Wet scrubber | - | 36 | 10.5 | - | 5950 | - | 55 | 10 | 5 | ND | ND |
| 19 | Induction | Wet scrubber | - | 33 | 10 | - | 3500 | - | 50 | 21 | 11 | ND | ND |
| 20 | Induction | Wet scrubber | - | 28 | 9.1 | - | 5120 | - | 48 | 18 | 10 | ND | ND |
| 21 | Induction | Cartridge filter | 200 | <50 | 9 | 40000 | 5185 | | <20 | ND | ND | ND | ND |

concentration data were available before the installation of the wet plug manifold. The current concentration that occurred after the installation of the wet plug collector in iron foundries 4, 6, 8, and 10 was 110 mg/Nm³, 102 mg/Nm³, 95 mg/Nm³, and 105 mg/Nm³, respectively. These values are within the emission limit of 150 mg/Nm³ [4–9].

## Impact of cyclone

As shown in Table 2, the cyclone is installed in iron foundry 1. A cyclone is a device used to filter particles with a diameter ranging from 1 to 1000 micrometres from a gas or liquid stream. The centrifugal force throws solid particles against the outer wall of the cyclone. After their fall, they are collected and separated in the deposition chamber. Table 1 shows that when the cyclone was used in an iron foundry, the SPM concentration reduced from 300 mg/Nm³ to 67 mg/Nm³. It was clear that the cyclone would be a suitable PCD to be installed in the dome to reduce contamination [4–9].

## Impact of wet scrubber

As shown in Table 2, the wet scrubber is installed in iron foundries 13, 14, 15, 16, 17, 18, 19, and 20. The wet scrubber is a term used to describe a variety. Devices that use a liquid to remove pollutants. In a wet scrubber, the contaminated gas stream comes into contact with

the washing liquid. The particles of the contaminated gas stream are collected by the liquid droplets. This phenomenon is achieved by dissolving or absorbing the particles in the liquid. Any droplets in the flue gases must then be separated from the clean exhaust stream using a device called a mist separator.

Most foundries have used wet scrubbers. The wet scrubber is very effective in removing both particulate matter and gases. During the study reported here, three iron foundries equipped with a Venturi wet scrubber and eight foundries equipped with a normal wet scrubber of different types (see Table 1) were followed. Interestingly, a remarkable reduction in particle emission concentrations was achieved in the Iron 17 foundries and the Iron 20 foundries. The SPM concentrations were found to be 52, 51, 50, 51, 55, 50 mg / Nm$^3$ in iron-13 foundries, iron-14 foundries, iron-15 foundries, iron-16 foundries, iron-18 foundries, and iron-19 foundries, respectively. These values show that the reduced particle emission magnitude obtained after installation of the wet scrubber is only 50 mg/nm [3, 4, 9].

## Impact of venturi wet scrubber

As shown in Table 2, the wet venturi scrubber is installed in iron foundries 5, 11, and 12. A wet venturi scrubber is essentially a channel with a converging venturi-shaped groove followed by a diverging cross-section. They are leading a stream of contaminated gas from the dome. The liquid is usually introduced in the form of jets, which atomize rapidly to form many tiny droplets [36]. The wash liquid is injected perpendicular to the incoming gas stream, breaking the liquid down into small droplets that are then used to collect particulate and gaseous pollutants. The energy required for diffusion is provided by a high-velocity gas stream. The quality of the liquid dispersion depends on the speed of the feed gas. A decrease in the gas velocity will result in a decrease in collection efficiency. A venturi wet scrubber has high separation efficiency for removing contaminants from the air. Table 1 shows that when the Venturi wet scrubber was installed in the dome, the resulting reduction in SPM concentration was 51 mg/Nm$^3$ in the iron foundry 5. When the venturi wet scrubber was installed in an induction furnace, the observed SPM concentrations were only 31 mg/Nm$^3$ and 37 mg/Nm$^3$, respectively, in iron foundries 11 and 12. Therefore, the control of the contamination depends on the nature of the SPM expelled from the stack and the type of furnace being used [4, 9].

## Impact of cartridge filter

As shown in Table 2, the cartridge filter is installed in foundry 21. It is pretty interesting to note that the use of the cartridge filter to control fouling from iron foundries has not yet been addressed by researchers. This is evidenced by the lack of papers stating that cartridge filters are installed in foundries. The cartridge filter equipped with an induction furnace in the iron foundry 21 reduced the concentration of SPM to less than 20 mg/Nm$^3$. This is the highest reduction in SPM concentration achieved as compared to that obtained by other PCDs discussed in the previous subsections.

The literature and field studies reported so far have revealed two facts. According to the first fact, the concentration of pollution caused by the induction furnace is lower than that caused by the cupola. The second fact is that of all PCDs, and the cartridge filter is the most effective in controlling fouling from the iron foundry. While the construction, operation, and performance of the induction furnace are well documented, this is not the case with the cartridge filter. Therefore, the construction, operation, and performance of the cartridge filter are described in the following section.

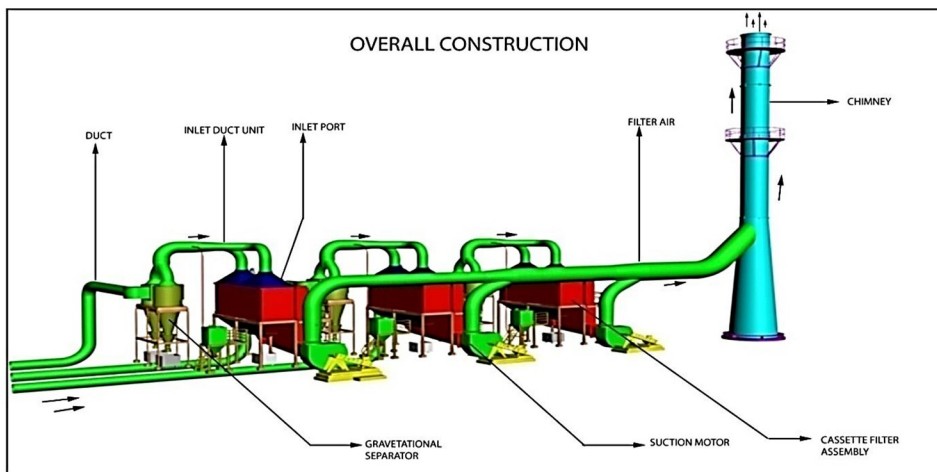

**Fig 2. Schematic layout of cartridge type filter assembly.**

## Construction and working principle of cartridge filter

Information regarding the construction, operation, and performance of the cartridge filter is collected from Foundry 21. This information is presented in this section. The schematic arrangement in which three sets of cartridge filters are used to control fouling in a foundry is shown in Fig 2. The detailed view of the cartridge filter is shown in Fig 3. As illustrated, flue gases enter from induction furnaces into the gravity separator through the inlet channels. Gravity separators separate heavy particles from the vapours. Energy Scrubber in Iron Foundry Industries is illustrated in Fig 4.

These heavy particles pass through cartridge filter assemblies. In the cartridge filter assembly, heavy particles are screened with a flat bag, and light particles of smaller sizes are released into the atmosphere. In this way, pollution from combustion gases is reduced. Compared to other PCDs, the cartridge filter consumes less energy to control contamination from iron

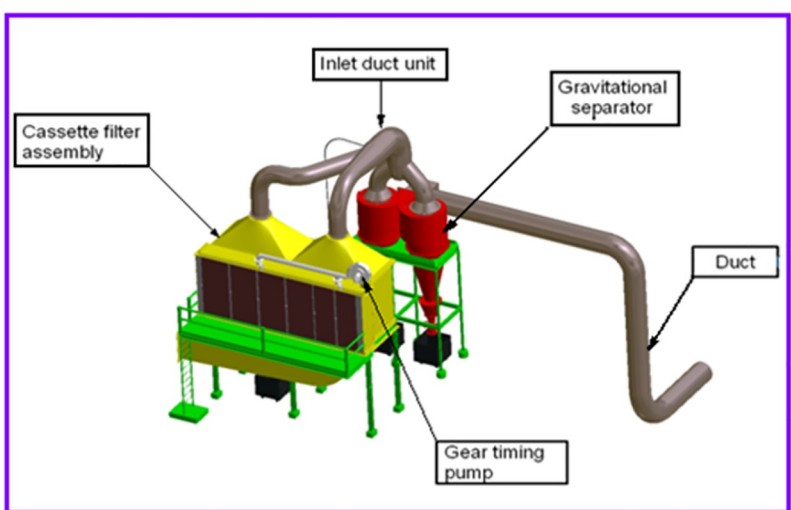

**Fig 3. The detailed view of the cartridge filter.**

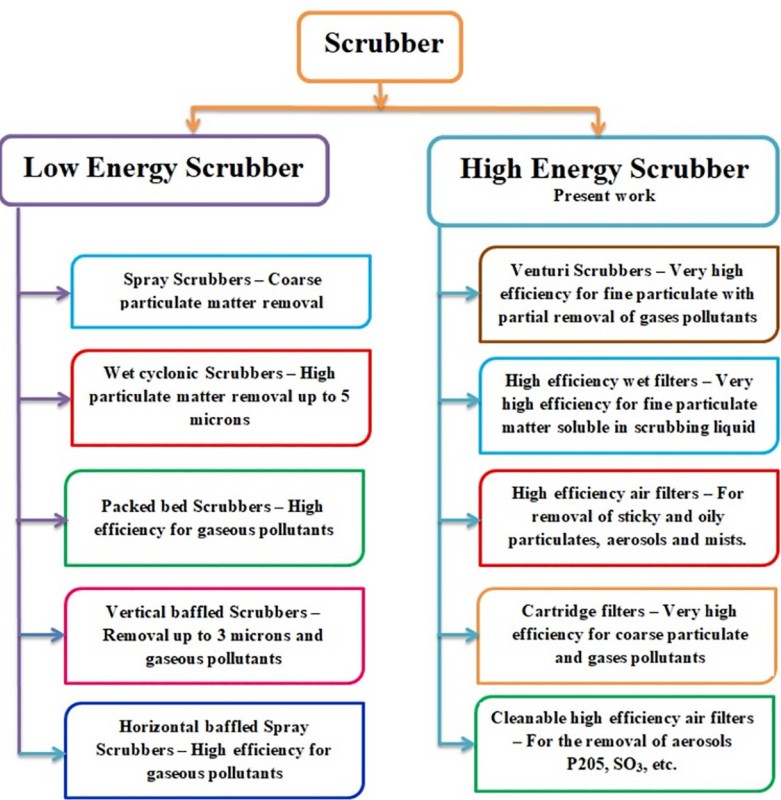

**Fig 4. Energy scrubber in iron foundry industries.**

foundries. The construction of the cartridge filter assembly is so simple that it allows easy cleaning of the inner chamber and replacement of the flat bag. When installed in an induction furnace, the wet scrubber is known to generate secondary pollution. On the other hand, there is no possibility for the cartridge filter to generate secondary pollution during use. During the study reported here, in the iron 21 foundries using a cartridge filter, the contamination level was found to be less than 20 mg/Nm$^3$. This pollution level is the standard set in European regulations. A minor drawback of using the cartridge filter is that it requires technically trained personnel to operate it. This minor inconvenience can be overcome by providing professional training to iron foundry operators to operate and maintain the cartridge filter.

## Conclusion

Foundries have been manufacturing products necessary to improve the quality of human life since ancient times [4, 9]. Many advances have been made over the years in foundry technologies and methods to produce high-quality and productive products. These developments have fueled the growth of foundries. Its growth has benefited several companies around the world. Iron foundries, in particular, have played an important role in increasing the prosperity of various countries and regions. On the other hand, the development of iron foundries is being felt in different parts of the world as the flue gases emitted by foundries affect the environment, and therefore the healthy life of humans is affected [37, 38]. To compensate for this shortage, some PCDs are used in iron foundries. Some studies conducted by the research community in this direction have shown that the installation of PCDs and pollution prevention from

foundries are influenced and influenced by local factors such as high costs, legal laws, lack of manpower, and skilled workforce. While research continues in this direction, it is surprising that the performance of Tamilnadu steel foundries from an environmental pollution point of view has not yet been reported in the literature. To fill this research gap, the research reported in this article was conducted.

After a literature review, relevant data were collected from foundries in the Tamilnadu State. The data collected indicates that the levels of contamination released by the induction furnace installed in foundries are lower than those in foundries that have installed domes. This derivation is confirmed by the guidelines of [30]. Another interesting observation is that a cartridge filter is used in only one foundry. The emission control level at this foundries was the highest compared to the installation of other PCDs. Details about the construction, operation, and performance of the cartridge filter are noticeably absent from the literature. That is why the details have been collected at the iron foundry in which the cartridge filter is equipped with an induction furnace. These details are presented in this article. A preliminary study showed that it is cheaper to install a cartridge filter than to install other PCDs in iron foundries. It is also striking that the maintenance of the cartridge filter in iron foundries is easier. The efficiency of the cartridge filter is so high that the SPM concentration released by it is lower than the standard value prescribed by European regulations. Amid all these advantages, the factors that can deter iron foundries from installing a cartridge filter need to be studied in more detail. At this point, this article concludes with an evaluation of the need to study iron foundries and examine the practice of using an induction furnace and a cartridge filter to reduce fouling and, thereby, the power generation capabilities of iron foundries.

## Supporting information

**S1 File.**
(DOCX)

## Author Contributions

**Conceptualization:** Krishnaraj Ramaswamy, Leta Tesfaye Jule, Nagaprasad N., Shanmugam. R., Priyanka Dwarampudi. L.

**Data curation:** Krishnaraj Ramaswamy, Leta Tesfaye Jule, Nagaprasad N., Kumaran Subramanian, Shanmugam. R., Priyanka Dwarampudi. L.

**Formal analysis:** Krishnaraj Ramaswamy, Leta Tesfaye Jule, Nagaprasad N., Kumaran Subramanian, Shanmugam. R., Priyanka Dwarampudi. L., Venkatesh Seenivasan.

**Investigation:** Krishnaraj Ramaswamy, Leta Tesfaye Jule, Nagaprasad N., Shanmugam. R., Priyanka Dwarampudi. L.

**Methodology:** Krishnaraj Ramaswamy, Leta Tesfaye Jule, Nagaprasad N., Shanmugam. R., Priyanka Dwarampudi. L.

**Software:** Krishnaraj Ramaswamy, Leta Tesfaye Jule, Shanmugam. R., Priyanka Dwarampudi. L.

**Supervision:** Krishnaraj Ramaswamy.

**Validation:** Krishnaraj Ramaswamy, Leta Tesfaye Jule, Nagaprasad N., Shanmugam. R., Priyanka Dwarampudi. L., Venkatesh Seenivasan.

**Writing – original draft:** Krishnaraj Ramaswamy.

**Writing – review & editing:** Krishnaraj Ramaswamy, Leta Tesfaye Jule, Nagaprasad N., Shanmugam. R., Priyanka Dwarampudi. L.

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
