## [Decision Letter · Decision Letter 0]

11 May 2022

PONE-D-22-10422Investigation on Pollution Control Device (PCD) in Foundry Industry to Reduce Environmental ChemicalsPLOS ONE

Dear Dr. Krishnaraj Ramaswamy,

Thank you for submitting your manuscript to PLOS ONE. After careful consideration, we feel that it has merit but does not fully meet PLOS ONE’s publication criteria as it currently stands. Therefore, we invite you to submit a revised version of the manuscript that addresses the points raised during the review process.

We look forward to receiving your revised manuscript.

Kind regards,

Sivasankar Palaniappan, Ph.D.,

Academic Editor

PLOS ONE

Journal Requirements:

"The funders had no role in study design, data collection and analysis, decision to publish, or preparation of the manuscript"

Additional Editor Comments:

Hello Krishnaraj Ramaswamy,

There were some concerns raised, Please see the attached reviewer comments for further details about necessary revisions. 

Reviewers' comments:

Reviewer's Responses to Questions

**Comments to the Author**

1. Is the manuscript technically sound, and do the data support the conclusions?

Reviewer #1: Partly

Reviewer #2: Yes

2. Has the statistical analysis been performed appropriately and rigorously? 

Reviewer #1: No

Reviewer #2: No

3. Have the authors made all data underlying the findings in their manuscript fully available?

Reviewer #1: Yes

Reviewer #2: No

4. Is the manuscript presented in an intelligible fashion and written in standard English?

Reviewer #1: Yes

Reviewer #2: Yes

5. Review Comments to the Author

Reviewer #1: The manuscript is not research article or a review article, both mixed to together and confuses the reader

The abstract not mentioning the results obtained from authors work. I feel it is lack of confidence.

Entire manuscript flow is few para review and another few what to be done but author failed to impress based on the work what they have done experimentally. The cartridge figure is computer design diagram, that create the doubt whether this work is done on the foundry or not. rewrite the manuscript with proper results and discussion and methods used to conclude the results with references. No error data present.

Reviewer #2: The manuscript titled “Investigation on Pollution Control Device (PCD) in Foundry Industry to Reduce Environmental Chemicals” written by authors is a good informative work in the field and useful for future researchers. The article ends by emphasizing the need to conduct surveys in foundries in which a cartridge filter is installed. The results of this study will provide useful information on the use of cartridge filters in induction furnaces to reduce foundry emissions. Though the results seem to be convincing there are some issues to be resolved in this manuscript. Addressing the below comments, I recommend the article to be published in the journal.

Recommendation: Major Revision

Comments:

1. In the abstract section, include the result of cartridge filter equipped with an Induction furnace.

2. In the result and discussion section compare the application of low energy scrubbers and High scrubbers from literature.

3. Check the units throughout the manuscript.

4. Since the authors has concentrated in Iron Foundry Industry. Hence the title has to revise as “Investigation on Pollution Control Device (PCD) in Iron Foundry Industry to Reduce Environmental chemicals.

5. What are the other industries which use the pollution control devices?

6. In general the manuscript needs editing in terms of language and choice of words for example in some places the word smelters was used. Instead of foundries. The clarity of the manuscript should be improved.

6. PLOS authors have the option to publish the peer review history of their article (what does this mean?). If published, this will include your full peer review and any attached files.

Reviewer #1: **Yes: **Sendilvelan Subramanian

Reviewer #2: No

---

## [Author Response · Author response to Decision Letter 0]

20 May 2022

Dear Editor,

Coauthors and I very much appreciated the encouraging, critical and constructive comments on this manuscript by the reviewer. The comments have been very thorough and useful in improving the manuscript. We strongly believe that the comments and suggestions have increased the scientific value of the revised manuscript by many folds. We have taken them fully into account in the revision. We are submitting the corrected manuscript with the suggestion incorporated into the manuscript (It is highlighted in red colour). The manuscript has been revised as per the comments given by the reviewer, and our responses to all the comments are as follows:

---

## [Decision Letter · Decision Letter 1]

23 Jun 2022

Investigation on Pollution Control Device (PCD) in Iron Foundry Industry to Reduce Environmental Chemicals

PONE-D-22-10422R1

Dear Dr. Krishnaraj Ramaswamy,

We’re pleased to inform you that your manuscript has been judged scientifically suitable for publication and will be formally accepted for publication once it meets all outstanding technical requirements.

Kind regards,

Sivasankar Palaniappan, Ph.D.,

Academic Editor

PLOS ONE

Additional Editor Comments (optional):

Reviewers' comments:

Reviewer's Responses to Questions

**Comments to the Author**

1. If the authors have adequately addressed your comments raised in a previous round of review and you feel that this manuscript is now acceptable for publication, you may indicate that here to bypass the “Comments to the Author” section, enter your conflict of interest statement in the “Confidential to Editor” section, and submit your "Accept" recommendation.

Reviewer #1: All comments have been addressed

Reviewer #2: All comments have been addressed

2. Is the manuscript technically sound, and do the data support the conclusions?

Reviewer #1: Yes

Reviewer #2: Yes

3. Has the statistical analysis been performed appropriately and rigorously? 

Reviewer #1: Yes

Reviewer #2: Yes

4. Have the authors made all data underlying the findings in their manuscript fully available?

Reviewer #1: Yes

Reviewer #2: Yes

5. Is the manuscript presented in an intelligible fashion and written in standard English?

Reviewer #1: Yes

Reviewer #2: Yes

6. Review Comments to the Author

Reviewer #1: all questions answered and modified in the manuscript appropriately. Although relatively less number of researches on controlling pollution is reported in

the literature arena, the construction and working of cartridge filters are not available

in the literature

Reviewer #2: (No Response)

7. PLOS authors have the option to publish the peer review history of their article (what does this mean?). If published, this will include your full peer review and any attached files.

Reviewer #1: **Yes: **Sendilvelan Subramanian

Reviewer #2: **Yes: **Narmadha MAnoranjan

---

## [Editor Report · Acceptance letter]

15 Jul 2022

PONE-D-22-10422R1 

Investigation on Pollution Control Device (PCD) in Iron Foundry Industry to Reduce Environmental Chemicals 

Dear Dr. Ramaswamy:

I'm pleased to inform you that your manuscript has been deemed suitable for publication in PLOS ONE. Congratulations! Your manuscript is now with our production department. 

Kind regards, 

on behalf of

Dr. Sivasankar Palaniappan 

Academic Editor

PLOS ONE